# Increased Immunoglobulin Gamma-3 Chain C in the Serum, Saliva, and Urine of Patients with Systemic Lupus Erythematosus

**DOI:** 10.3390/ijms24086927

**Published:** 2023-04-08

**Authors:** Ju-Yang Jung, Ji-Won Kim, Sang-Won Lee, Wook-Young Baek, Hyoun-Ah Kim, Chang-Hee Suh

**Affiliations:** 1Department of Rheumatology, Ajou University School of Medicine, Suwon 16499, Republic of Korea; 2Department of Molecular Science and Technology, Ajou University, Suwon 16499, Republic of Korea

**Keywords:** systemic lupus erythematosus, immunoglobulin G, biomarker, serum, saliva, urine

## Abstract

Immunoglobulin gamma-3 chain C (IGHG3) levels have been detected in the blood and tissue of patients with systemic lupus erythematosus (SLE). This study aims to assess its clinical value by measuring and comparing levels of IGHG3 in different body fluids in patients with SLE. The levels of IGHG3 in saliva, serum, and urine from 181 patients with SLE and 99 healthy controls were measured and analyzed. In patients with SLE and healthy controls, salivary IGHG3 levels were 3078.9 ± 2473.8 and 1413.6 ± 1075.3 ng/mL, serum IGHG3 levels were 478.1 ± 160.9 and 364.4 ± 97.9 μg/mL, and urine IGHG3 levels were 64.0 ± 74.5 and 27.1 ± 16.2 ng/mL, respectively (all *p* < 0.001). Salivary IGHG3 was correlated with ESR (correlation coefficient [r], 0.173; *p* = 0.024). Serum IGHG3 was correlated with leukocyte count (r, −0.219; *p* = 0.003), lymphocyte count (r, 0.22; *p* = 0.03), anti-dsDNA antibody positivity (r, 0.22; *p* = 0.003), and C3 levels (r, −0.23; *p* = 0.002). Urinary IGHG3 was correlated with hemoglobin level (r, −0.183; *p* = 0.021), ESR (r, 0.204; *p* = 0.01), anti-dsDNA antibody positivity (r, 0.262; *p* = 0.001), C3 levels (r, −0.202; *p* = 0.011), and SLE disease activity index (r, 0.332; *p* = 0.01). Urinary IGHG3 was higher in patients with nephritis than in those without (119.5 ± 110.0 vs. 49.8 ± 54.4 ng/mL; *p* < 0.01). IGHG3 was increased in the saliva, serum, and urine of patients with SLE. While salivary IGHG3 was not identified to be specific to SLE disease activity, serum IGHG3 showed correlations with clinical characteristics. Urinary IGHG3 levels were associated with disease activity and renal involvement in SLE.

## 1. Introduction

Systemic lupus erythematosus (SLE) is an autoimmune disease characterized by diverse manifestations including skin rash, arthritis, and nephritis, and immunological deterioration including activation of immune cells, loss of self-tolerance, and production of autoantibodies [1]. Inflammatory responses resulting from such immunopathological mechanisms repeatedly occur in target tissues, leading to organ damage or permanent dysfunction [2]. The manifestations of SLE vary according to the organs involved, and disease severity also changes with time or treatment. Therefore, regular monitoring of manifestations and disease activities is essential in the management of SLE. The most used tools for monitoring disease status include clinical symptoms and laboratory biomarkers for lupus disease activity. Although there are several laboratory biomarkers, including anti-double-stranded deoxyribonucleic acid (dsDNA) antibodies and complement levels (C3 and C4), they have some limitations [3]. Some patients show normal values, and these biomarkers do not precisely represent changes in disease activity in a considerable number of patients. Further, organ involvement cannot be assessed by these biomarkers.

Previously, we looked for the alteration of protein composition in saliva in patients with SLE to find a salivary biomarker for SLE and identified that several peptides including immunoglobulin gamma-3 chain C region (IGHG3) were elevated in the saliva of patients with SLE [4]. IGHG3 is immunoglobulin G (IgG)3, one of the subclasses of IgG. Most subclasses of IgG3 activate proinflammatory signals through Fc portions of IgG molecules (FcγRs) on immune cells [5]. IgG3 is composed of 5–8% IgG in humans and is characterized by an elongated hinge region, greater flexibility, and additional glycosylation sites. Due to such structural advantages, IgG3 acts as a potent immunoglobulin, leading to enhanced effector functions including complement activation, antibody-mediated phagocytosis, antibody-dependent cellular cytotoxicity, and interferon (IFN) production [6,7]. Complement activation and release of type I IFN through IgG-FcγRs by immune cells had been found to take a major role in the pathogenesis of SLE [8,9]. Hypergammaglobulinemia is a common manifestation and IgG-immune complex deposition in target tissues results in organ damage in SLE. However, the distribution of the IgG subclass in the sera and tissues and their respective functions have not been clearly elucidated in lupus. Previously, our data showed salivary IGHG3 levels were higher and associated with disease activity markers in patients with SLE [4].

Therefore, in the present study, we aimed to compare IGHG3 levels in different body fluids, saliva, serum, and urine of patients with SLE and their utility for diagnostic biomarkers for SLE. In addition, we aimed to analyze correlations between salivary, serum, and urine IGHG3 levels and disease activity markers or manifestations in patients with SLE.

## 2. Results

### 2.1. Basic Characteristics of Patients

The mean age of the patients with SLE donating saliva, serum, and urine was 38.8 ± 10.3, 38.9 ± 9.9, and 41.1 ± 10.5 years, respectively, and age and sex were matched with healthy controls (HCs) (Table 1). Mean disease duration was 6.6 ± 5.3, 6.9 ± 5.1, and 7.89 ± 5.6 years in the saliva, serum, and urine group, respectively, and anti-dsDNA antibody positivity was found in 36.7%, 41.0, and 38.2% of the saliva, serum, and urine group, respectively. In manifestations of SLE, 41.4%, 44%, and 30.6% of patients had arthritis, and 17.8%, 19.4%, and 21.0% had active lupus nephritis (LN) in the saliva, serum, and urine group, respectively. In addition, 21.2%, 26.1%, and 14.0% had hematologic involvement, respectively, and the mean SLEDAI score was 3.9 ± 3.4, 4.3 ± 3.3, and 4.1 ± 3.9 in the saliva, serum, and urine group, respectively.

### 2.2. Comparison of IGHG3 in Saliva, Serum, and Urine of Patients with SLE

In patients with SLE and HCs, the mean salivary IGHG3 level was 3078.9 ± 2473.8 and 1413.6 ± 1075.3 ng/mL, respectively (*p* < 0.001), the mean serum IGHG3 level was 478.1 ± 16.09 and 364.4 ± 97.9 μg/mL, respectively (*p* < 0.001), and the mean urinary IGHG3 level was 64.0 ± 74.5 and 27. ± 16.2 ng/mL, respectively (*p* < 0.001) (Figure 1). IGHG3 levels were significantly higher in the saliva, serum, and urine of patients with SLE compared with those of HCs.

### 2.3. ROC Curve of IGHG3 Levels for Discriminating SLE

The receiver operating characteristic (ROC) curves for salivary, serum, and urine IGHG3 levels for discriminating SLE are shown in Figure 2. The area under the curve (AUC) for salivary, serum, and urine IGHG3 levels was 0.734 (95% confidence interval [CI] 0.675–0.794), 0.823 (95% CI 0.673–0.794), and 0.733 (95% CI 0.67–0.795), respectively. The optimal cutoff values for salivary, serum, and urine IGHG3 were 1827.9 ng/mL, 345.6 μg/mL, and 34.2 ng/mL, respectively. Regarding the diagnostic ability characteristics of the biomarkers, the sensitivity of salivary, serum, and urinary IGHG3 levels was 0.598%, 0.823%, and 0.643%, respectively, and the specificity of IGHG3 was 0.768%, 0.542%, and 0.75%, respectively. According to the cutoff value for salivary, serum, and urinary IGHG3 levels, the positive predictive value (PPV) was 81.5%, 77.2%, and 82.1%, respectively, and the and negative predictive value (NPV) was 52.8%, 61.9%, and 54.1%, respectively. The comparison of ROC curves identified that the AUC of IGHG3 levels in serum was better in distinguishing SLE compared with those in saliva and urine.

### 2.4. Correlation between IGHG3 Levels and Clinical Markers in SLE

The IGHG3 levels in each type of body fluid were analyzed to determine their associations with laboratory findings and SLE disease activity index (SLEDAI)-2K (Table 2). The salivary IGHG3 levels correlated with the erythrocyte sedimentation rate (ESR) (correlation coefficient [r], 0.173, *p* = 0.024) in patients with SLE. The serum IGHG3 levels negatively correlated with the leukocyte count (r, −0.219, *p* = 0.003), lymphocyte count (r, −0.221, *p* = 0.03), and C3 level (r, −0.23, *p* = 0.002) and positively correlated with anti-dsDNA antibody positivity (r, 0.22, *p* = 0.003) and SLEDAI (r, 0.147, *p* = 0.048). The urinary IGHG3 level negatively correlated with the hemoglobin level (r, −0.183, *p* = 0.021) and C3 level (r, −0.202, *p* = 0.011) and positively correlated with the ESR (r, 0.204, *p* = 0.01), anti-dsDNA antibody positivity (r, 0.262, *p* = 0.001), and SLEDAI-2K (r, 0.332, *p* < 0.01).

### 2.5. Comparisons of IGHG3 Levels between Patients with Specific Manifestation and Those Not

Salivary IGHG3 levels did not differ between the two groups categorized according to the presence and absence of specific manifestations of SLE (Table 3). Serum IGHG3 levels were significantly lower in patients with fever than in those without fever (389 [260, 679.2] vs. 449.5 [153.8, 1062.3] μg/mL, *p* = 0.049). Urinary IGHG3 levels were significantly higher in patients with active LN (92.5 [24.5, 502.8] vs. 36.4 [3.3, 437.1] ng/mL, *p* < 0.001), and in patients with SLEDAI-2K > 6 (76.7 [24.5, 502.8] vs. 37.3 [3.3, 437.1] ng/mL, *p* < 0.001) than in those with SLEDAI-2K < 6.

### 2.6. ROC Curve for Urinary IGHG3 Levels for Discriminating Active Lupus Nephritis

The ROC curve for urinary IGHG3 for discriminating active LN is shown in Figure 3. The AUC was 0.816 (95% CI 0.738–0.893) and the optimal cutoff value was 56.7 ng/mL. The PPV and NPV were 46.3% and 93.2%, respectively. Regarding diagnostic ability characteristics as biomarkers, the sensitivity and specificity of urinary IGHG3 were 78.1% and 76.8%, respectively.

## 3. Discussion

In this study, salivary, serum, and urinary IGHG3 levels were significantly higher in patients with SLE than in HCs. Salivary IGHG3 levels correlated with the ESR. Serum IGHG3 levels negatively correlated with leukocyte and lymphocyte counts and C3 levels, and positively correlated with anti-dsDNA antibody positivity. Urinary IGHG3 levels positively correlated with the ESR, anti-dsDNA antibody positivity, and SLEDAI-2K score and negatively correlated with hemoglobin and C3 levels. In addition, urinary IGHG3 levels were increased in patients with active LN or SLEDAI-2K score >6 and showed significant efficacy in discriminating active LN in patients with SLE.

IgG3 has an elongated hinge of 62 amino acids with a polyproline helix and 11 disulfide bridges, which enables the activation of FcγR-mediated effector functions and the complement cascade [10,11]. Due to its length and flexibility, IgG3 has a unique role in protecting against encapsulated polysaccharides through greater functional affinity and binding [12,13]. IgG3 binds to C1q on the viral surface, leading to intracellular neutralization via the C1/C4 pathway [14]. C1q on cellular debris is well-known to recruit autoimmune response in SLE. While IgG2 produces T helper-17 cell-inducing cytokines, such as tumor necrosis factor, interleukin (IL)-1β, and IL-23, IgG3 modulates the type I IFN response, which is critical in SLE [15]. In addition, IFN-γ stimulates the expression of IgG3 on B cells through a T cell-independent immune response, and SLE had been found to induce an IFNγ-rich environment [16]. In mice with lupus phenotype mice (Fcgr2b-cKO mice), marginal zone B cells, B-1 cells, and plasma cells produce higher amounts of IgG3 than other subclasses [17]. These data suggest that IgG3 might be involved in the pathogenesis of SLE.

There are several studies on IgG3 levels in the blood or IgG3 deposition in the tissues of patients with SLE. A study reported that IgG1, IgG2, and IgG3 levels in the serum were elevated in patients with SLE compared with those in HCs, and the IgG3 level decreased in patients in remission [18]. IgG3 was predominant in 13 among 20 skin tissues from patients with SLE, which showed IgG-positive lupus bands [19]. The antinucleosome IgG3 levels were higher in patients with active SLE than in those with other autoimmune diseases or inactive SLE [20]. In addition, antinucleosome IgG3 levels correlated with SLEDAI scores and were elevated in patients with active SLE with nephritis compared with those without nephritis; however, anti-dsDNA IgG3 levels did not. In another study, not only IgG3 but also IgG1 and IgG2 were elevated in patients with SLE [21]. In this study, serum IGHG3 levels were higher, negatively correlated with leukocyte and lymphocyte counts and C3 levels, and positively correlated with anti-dsDNA antibody and SLEDAI-2K score. As laboratory markers, low C3 and anti-dsDNA antibody positivity indicate higher SLE activity, and leukopenia and lymphopenia indicate hematologic involvement of SLE, suggesting that serum IGHG3 levels are correlated with lupus activity. However, the AUC of serum IGHG3 levels was poorer than that of salivary and urinary IGHG3, and serum IGHG3 levels did not differ between the two groups categorized according to the presence and absence of specific manifestations of SLE. Serum IGHG3 levels were lower in patients with fever than in those without, but the population who had a fever was too small to consider a statistical association.

Previously, our data showed that IgG3 levels in the saliva were higher in patients with SLE than in HCs and patients with rheumatoid arthritis; the levels also correlated with the ESR, anti-dsDNA antibody positivity, and presence of active nephritis [4]. In the present study, salivary IgG3 levels were correlated with the ESR, but not with anti-dsDNA antibody positivity or the presence of nephritis. The ESR indicates a chronic inflammatory condition in infectious and autoimmune diseases; therefore, IgG3 levels might be increased in the saliva during systemic prolonged systemic inflammatory status. There may be a unique mechanism involving enhanced secretion of IgG3 through the salivary gland. However, further studies are needed to clarify this.

Clinically, active LN is identified by histological lesions defined as class III, IV, or V according to the ISN/RPS 2003 classification. LN presents with active inflammation and shows a poor prognosis without treatment [22]. Selective inhibition of IgG3 production attenuated renal disease in mice of MRL/lpr or other strains prone to autoimmune phenotypes [23]. IgG3-deficient mice with an MRL/lpr genetic background had less extensive glomerulosclerosis and lived longer than those with IgG3 [24]. IgG3 depletion by early thymic irradiation ameliorates nephritis and enhanced survival in NZM/W mice [23]. While IgG2 antinucleohistone and IgG1 anti-dsDNA antibodies were raised in patients with renal flares compared with patients with extrarenal flares, IgG3 did not rise [25]. In a study with human renal tissue, IgG2 and IgG3 were highly deposited in the capillary loops of patients with class IV and V LN compared with patients with class I, II, and III LN who exhibited much less deposition (*p* < 0.01); however, this trend was not observed for IgG1 and 4 [26]. IgG3 deposition was predominant in 38% of patients with class IV LN, and renal survival was shorter in patients with renal tissues with IgG3 deposition compared with patients with renal tissues without IgG3 deposition. IgG3 may promote active inflammation through the renal capillary loop by binding to FcγRs and C1q, leading to the activation of immune cells, including dendritic cells and macrophages, and immune complex deposition [27,28]. In this study, the urinary IgG3 level was significantly elevated in patients with active LN and showed a reliable AUC in the utility analysis. In addition to its pathological role as a stimulator of autoimmune response, urinary IgG3 may be a reliable tool for differentiating active nephritis among patients with SLE.

Urinary IgG3 levels correlated with the hemoglobin level, C3 level, ESR, anti-dsDNA antibody positivity, and SLEDAI-2K score, suggesting that higher urinary IgG3 levels indicate an active status of SLE or LN. Urine sampling is non-invasive and easier than blood sampling, which is mandatory in the management of SLE. Urinary IgG3 cannot be a substitute for several disease activity indicators in blood tests, but it might be helpful to differentiate exacerbation of SLE and other diseases in situations of fever or skin rash. Moreover, urinary IgG3 showed a remarkable value in diagnostic active LN. Although more research is needed, urinary IgG3 tests could be used in checking the presence of active LN. Many patients with active LN present with only asymptomatic proteinuria, resulting in delayed intervention and organ damage. It might be considered beneficial to add IgG3 level to proteinuria in regular urinalysis.

This study has some limitations. IgG3 concentrations in each type of body fluid could not be compared in the same patients with SLE because several patients failed or refused to provide specific samples such as saliva or urine. In addition, urinary IgG3 concentrations could not be compared with the levels of patients with other nephropathies and other connective tissue disorders to assess as a biomarker for lupus nephritis. Most patients were treated using the standard strategy for SLE management; therefore, the effect of the maintained drugs could not be ignored in the body fluids. However, in clinical practice, monitoring of clinical biomarkers is conducted during management or follow-up, and the IGHG3 levels in the current study are representative of those in most patients with SLE. In addition, this is the first attempt to compare IGHG3 levels in different body fluids of a relatively large number of patients with SLE.

Salivary, serum, and urinary IGHG3 levels were higher in patients with SLE than in HCs. Salivary IGHG3 correlated with the ESR. Serum IGHG3 levels correlated with leukocyte and lymphocyte counts, C3 levels, and anti-dsDNA antibody positivity. Urinary IGHG3 levels correlated with hemoglobin, C3 levels, the ESR, anti-dsDNA antibody positivity, and SLEDAI-2K score. Urinary IGHG3 levels were higher in patients with active nephritis or SLEDAI-2K score > 6. In conclusion, salivary, serum, and urinary IGHG3 levels were higher in patients with SLE. The measurement of urinary IGHG3 levels can help differentiate active nephritis.

## 4. Materials and Methods

### 4.1. Subjects and Data Collection

A total of 181 patients who met the 1997 update of the 1982 American College of Rheumatology criteria [29] or the 2012 Systemic Lupus International Collaborating Clinics classification criteria [30] for SLE and 99 age- and sex-matched HC participated in this study. No patients had malignancy, infection, or other autoimmune diseases, and all were enrolled in the rheumatology clinic of Ajou University Hospital.

At the time of enrollment, saliva, plasma, and urine samples were collected from all patients. Clinical data, including information on demographics, medical history, manifestations of SLE and non-SLE, laboratory findings, and treatment, were collected from medical records and through interviews. Laboratory findings included complete blood count; ESR; and levels of C3, C4, antinuclear antibodies (ANA), and anti-dsDNA antibodies. Anti-dsDNA antibody levels were measured using a commercial kit (Trinity Biotech, Bray, Ireland) with a normal range of <7 IU/mL. C3 and C4 levels were measured using Cobas (Roche Diagnostics, Basel, Switzerland) with normal ranges of 90–180 mg/dL and 10–40 mg/dL, respectively. In all the patients, active LN was confirmed based on histopathological results of renal tissue as class III, IV, or V based on the International Society of Nephrology/Renal Pathology Society (ISN/RPS) 2003 classification [22]. The SLEDAI-2K score was calculated using clinical, laboratory, and serologic features observed within the last 10 days from the time of enrollment. Comprehensive medication histories, including the use of glucocorticoids and immunosuppressants, were obtained.

### 4.2. Measurement of IGHG3 in Saliva, Serum, and Urine

After sample collection, the blood and urine were centrifuged at 15,928 relative centrifugal force for 10 min. There are two methods, the stimulated and unstimulated way, to collect saliva in clinical practice or research. Here, the saliva was collected using the unstimulated method, which is without the use of any stimulator. The sample was then centrifuged at 1763 RCF for 10 min. The supernatants were collected and stored at −80 °C until further analysis. IGHG3 levels in the saliva, serum, and urine were measured by enzyme-linked immunosorbent assay (ab 137981, Abcam, Cambridge, UK). All materials were supplied with the kit, and the test was performed according to the manufacturer’s instructions.

### 4.3. Statistical Analysis

The Mann–Whitney U test was performed to compare IGHG3 concentration in two groups, Spearman’s correlation coefficient was used in bivariate correlation to obtain a correlation coefficient (rho) between disease activity markers and IGHG3 concentrations in each body fluid, and multiple regression analysis was performed. By analyzing the AUC of the ROC curve, we established the utility of salivary, serum, and urinary IGHG3 levels. Youden’s index was used to determine the cutoff values and calculated sensitivity, specificity, PPV, and NPV. All statistical analyses were performed using SPSS software (version 25.0; IBM Corporation, Armonk, NY, USA), and statistical significance was set at *p* < 0.05.

## Figures and Tables

**Figure 1 ijms-24-06927-f001:**
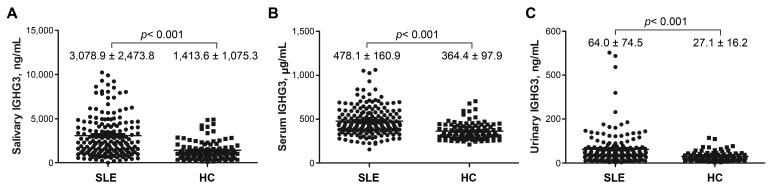
Immunoglobulin gamma-3 chain C (IGHG3) levels in patients with SLE and HCs. (**A**) Saliva, (**B**) serum, and (**C**) urine. SLE, systemic lupus erythematosus; HC, healthy controls.

**Figure 2 ijms-24-06927-f002:**
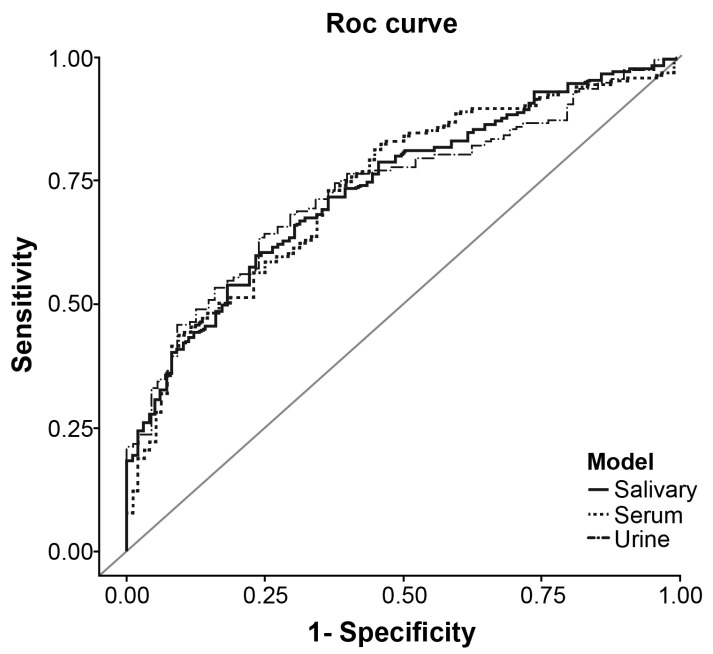
Receiver operating characteristic (ROC) curves associated with the diagnostic utility of IGHG3 levels in the saliva, serum, and urine. For systemic lupus erythematosus diagnosis, the area under the receiver operating characteristic curve was 0.734 (95% CI 0.675–0.794) for IGHG3 in the saliva, 0.733 (95% CI 0.673–0.794) for IGHG3 in the serum, and 0.733 (95% CI 0.67–0.795) for IGHG3 in the urine. SLE, systemic lupus erythematosus; HC, healthy controls.

**Figure 3 ijms-24-06927-f003:**
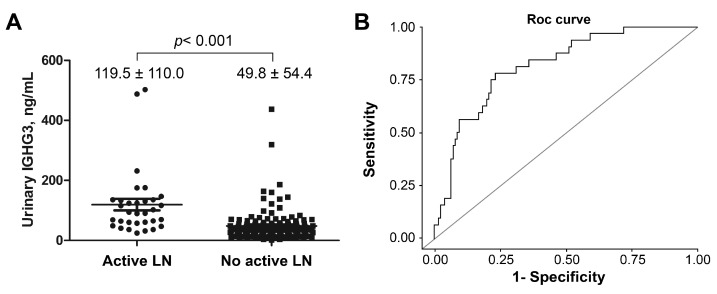
(**A**) Immunoglobulin gamma-3 chain C (IGHG3) levels in patients with and without active lupus nephritis (LN). (**B**) Receiver operating characteristic (ROC) curves associated with the diagnostic utility of urinary IGHG3 for active LN. The area under the receiver operating characteristic curve was 0.816 (95% CI 0.738–0.893).

**Table 1 ijms-24-06927-t001:** Basic characteristics of patients.

	Saliva	Serum	Urine
N (vs. HC)	169 (99)	181 (96)	157 (88)
Age, year (vs. HC) *	38.8 ± 10.3 (37.3 ± 9.0)	38.9 ± 9.9 (37.2 ± 9.0)	41.1 ± 10.5 (39.1 ± 8.2)
Sex, F/M (vs. HC) *	162:7 (94:5)	125:9 (88:8)	150:7 (84:4)
Disease duration, years	6.6 ± 5.3	6.9 ± 5.1	7.8 ± 5.6
WBC, /μL	4979.3 ± 2169.2	5034.6 ± 2372.7	4893.9 ± 2238.1
Hemoglobin, g/dL	13.6 ± 14.4	13.7 ± 16.1	12.5 ± 1.8
Platelet, ×10^3^/μL	226.4 ± 65.3	216.9 ± 67.9	229.8 ± 81.1
Lymphocyte, /μL	1456.2 ± 707.7	1366.6 ± 750.0	1391.9 ± 603.0
ESR, mm/h	16.9 ± 14.1	20.9 ± 21.4	15.7 ± 13.7
Anti-dsDNA (+), n (%)	62 (36.7)	55 (41.0)	60 (38.2)
C3, mg/dL	89.2 ± 23.7	87.8 ± 27.5	88.7 ± 24.5
C4, mg/dL	19.6 ± 10.2	19.4 ± 10.4	18.3 ± 9.0
Fever, n (%)	7 (4.2)	14 (7.7)	5 (3.2)
Oral ulcer, n (%)	22 (13.0)	21 (11.6)	18 (11.5)
Alopecia, n (%)	25 (14.8)	25 (13.8)	26 (16.6)
Skin rash, n (%)	38 (22.5)	26 (19.4)	33 (21.0)
Nephritis (ISN/RPS class III, IV, or V), n (%)	30 (17.8)	26 (19.4)	32 (20.4)
Arthritis, n (%)	70 (41.4)	59 (44.0)	48 (30.6)
Hematologic disease, n (%)	36 (21.3)	35 (26.1)	22 (14.0)
SLEDAI-2K	3.9 ± 3.4	4.3 ± 3.3	4.1 ± 3.9

HC, healthy controls; ESR, erythrocyte sedimentation rate; dsDNA, double-strand deoxyribonucleic acid; C, complement; ISN/RPS, International Society of Nephrology/Renal Pathology Society; SLEDAI, systemic lupus erythematosus disease activity index. * Matched.

**Table 2 ijms-24-06927-t002:** Correlation between disease activity markers and IGHG3 in patients with systemic lupus erythematosus.

Disease Activity Markers	Saliva IGHG3		Serum IGHG3		Urine IGHG3	
	r	*p*-Value	r	*p*-Value	r	*p*-Value
Disease duration	−0.073	0.346	−0.01	0.889	0.068	0.4
Leukocyte count	−0.138	0.075	−0.219	0.003	0.017	0.837
Hemoglobin	−0.015	0.85	0.058	0.439	−0.183	0.021
Platelet count	−0.117	0.131	−0.127	0.09	0.02	0.799
Lymphocyte count	−0.049	0.525	−0.221	0.03	−0.049	0.539
ESR	0.173	0.024	0.07	0.351	0.204	0.01
Anti-dsDNA Ab (+)	0.038	0.625	0.22	0.003	0.262	0.001
Complement 3	−0.122	0.114	−0.23	0.002	−0.202	0.011
Complement 4	−0.061	0.43	−0.146	0.05	−0.134	0.094
SLDEAI-2K	0.107	0.166	0.147	0.048	0.332	<0.001

IGHG3, immunoglobulin gamma 3 chain C region; r, correlation coefficient; ESR, erythrocyte sedimentation rate; dsDNA, double-strand deoxyribonucleic acid; SLEDAI, systemic lupus erythematosus disease activity index.

**Table 3 ijms-24-06927-t003:** Comparisons of IGHG3 levels in each body fluid between patients with specific manifestations and those without.

Manifestations	Saliva IGHG3, ng/mL		*p*-Value	Serum IGHG3, μg/mL		*p*-Value	Urine IGHG3, ng/mL		*p*-Value
Fever	(+)	1658.2 (690, 5034.6)	0.122	(+)	389 (260, 679.2)	0.049	(+)	51.4 (12.5, 130.6)	0.96
	(−)	2293 (20, 14,222)		(−)	449.5 (153.8, 1062.3)		(−)	43.1 (3.3, 502.8)	
Skin rash	(+)	2280.2 (417.2, 14,222)	0.314	(+)	437.4 (227.9, 1051.9)	0.58	(+)	36.2 (11.1, 437.1)	0.048
	(−)	2377.9 (20, 9895.7)		(−)	446 (153.8, 1062.3)		(−)	44.1 (3.3, 502.8)	
Oral ulcer	(+)	3248.3 (417.2, 8845.1)	0.351	(+)	430 (265, 833)	0.749	(+)	44.8 (10, 121.6)	0.859
	(−)	2274 (20, 14,222)		(−)	446 (153.8, 1062.3)		(−)	42.8 (3.3, 502.8)	
Alopecia	(+)	1610 (417.2, 6096.7)	0.168	(+)	489 (265, 839.9)	0.568	(+)	37.5 (9.1, 232)	0.726
	(−)	2386.9 (20, 14,222)		(−)	442.7 (153.8, 1062.3)		(−)	43.8 (3.3, 502.8)	
Arthritis	(+)	2147.4 (206, 14,222)	0.282	(+)	432.8 (227.9, 1062.3)	0.342	(+)	36.7 (3.3, 185.7)	0.123
	(−)	2480.8 (20, 10,217)		(−)	453 (153.8, 1025)		(−)	47.3 (6.8, 502.8)	
Class III/IV/V nephritis *	(+)	2088.6 (342, 9214.3)	0.813	(+)	469.1 (218.3, 1025)	0.48	(+)	92.5 (24.5, 502.8)	<0.001
	(−)	2296 (20, 14,222)		(−)	436.9 (153.8, 1062.3)		(−)	36.4 (3.3, 437.1)	
Hematologic involvement	(+)	2824.6 (206, 9296)	0.115	(+)	451.2 (241.2, 930)	0.365	(+)	46.4 (6.8, 319.2)	0.768
	(−)	2206 (20, 14,222)		(−)	441 (153.8, 1062.3)		(−)	42.6 (3.3, 502.8)	
SLEDAI−2K > 6	(+)	2045.8 (658, 14,222)	0.98	(+)	474.7 (218.3, 837.9)	0.29	(+)	76.7 (24.5, 502.8)	<0.001
	(−)	2296 (20, 10,217)		(−)	432.4 (153.8, 1062.3)		(−)	37.3 (3.3, 437.1)	

(+), level of patients who have the manifestation; (−), level of patients who do not have the manifestation; IGHG3, immunoglobulin gamma 3 chain C region; SLEDAI, systemic lupus erythematosus disease activity index. * By International Society of Nephrology/Renal Pathology Society (ISN/RPS) classification criteria for lupus nephritis.

## Data Availability

Not applicable.

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
