# Peer review of "Increased Immunoglobulin Gamma-3 Chain C in the Serum, Saliva, and Urine of Patients with Systemic Lupus Erythematosus"

_ijms, 2023, doi:10.3390/ijms24086927_

Round 1
Reviewer 1 Report
The authors present data on immunoglobulin gamma-3 Chain c (IGHG3) in patients with SLE. The aim of the study is to evaluate, if IGHG3 in serum, urine or saliva can be used as marker for SLE diagnosis and disease activity. The authors state, that IGHG3 is increased in all 3 fluids in SLE patients and that urinary IGHG3 levels are associated with disease activity and renal involvement. Really interesting paper on a potential activity marker.
The authors present a relatively large cohort of SLE patients. Also the study design is reasonable and solid.
Point by point:
Abstract:
The story can be described better in the abstract. The authors start directly with presenting data. I would suggest to present a 1-sentence-aim at the beginning and a conclusion in 1 sentence at the end.
lines 47-51: sentence is difficult to understand
lines 53-56: I do not understand the sentence. Also: You mention, that the distribution of IgG subtypes is not clearly elucidated and you state in line 57, that this is the reason to analyse IGHG3. For me, this argumentation is not conclusive. This raises the question why you only measured IgG3 and not IgG1, 2, 4....
line 64: please rewrite the sentence like this: "The mean (or median?) age of the patients donating saliva, serum and urine was .....". Otherwise it does not make much sense. also: please mention also the mean or median age of the healty controls. Regarding the question if mean or median should be displayed in this manuscript: Please test for normal distribution. But in the methods section you mention that you used Mann Whitney test for significance. So you assume that the data is not normal distributed. It would be better to show median and range for all the data...
line 88: ROC curves. Please describe in the method section how you produced the ROC curves. There are different methods to do this.
line 93: ng or µg? please use the same unit.
line 157-167: I am not sure, why this description of IgG3 structure and polysaccharide binding is relevant. As I understand it, it is used as a potential marker in the paper. The link to SLE is not clear. Can you describe more, why the structure and binding characteristics are important for SLE and is there an idea, why IGHG3 is elevated in SLE?
line 226: can you say how many patients you have who provided all 3 specimen or 2? Would it be interesting to show correlated data on this small cohort?
line 269: what is "the unstimulated method"? please explain briefly
Discussion: The value of this marker for diagnosing active nephritis is quite clear. Can you discuss, what benefit IGHG3 has over other parameters (C3 levels, anti-dsDNA, SLEDAI) for the assessment of disease activity and for diagnosis? How do you propose to use IGHG3 in the future?
Author Response
Dear Reviewer,
We appreciate your review of our manuscript “Immunoglobulin Gamma-3 Chain C in plasma, urine, and saliva in systemic lupus erythematosus”. In response to your comments and those of the reviewers, we have made several changes to the text, as summarized below.
Abstract:
- The story can be described better in the abstract. The authors start directly with presenting data. I would suggest presenting a1-sentence-aim at the beginning and a conclusion in 1 sentence at the end.
Answer) Thank you for the comment. We added the sentences as below and underlined them in the revised manuscript.
- This study aims to assess its clinical value through measuring and comparing levels of IGHG3 in plasma, saliva, and urine in patients with SLE.
- While salivary IGHG3 was not identified to be specific to SLE disease activity, serum IGHG3 showed correlations with clinical characteristics.
- lines 47-51: sentence is difficult to understand.
Answer) Thank you for the comment. We modified the sentences as below and underlined them in the revised manuscript.
- IGHG3 is immunoglobulin G (IgG)3, one of subclasses of IgG. Most subclasses of IgG3 activate proinflammatory signals through Fc portions of IgG molecules (FcγRs) on immune cells. IgG3 is composed 5-8% of IgG in human and characterized by an elongated hinge region, greater flexibility, and additional glycosylation sites. By such structural advantages, IgG3 acts as a potent immunoglobulin, leading to enhanced effector functions including complement activation, antibody-mediated phagocytosis, antibody-dependent cellular cytotoxicity, interferon (IFN) production.
- lines 53-56: I do not understand the sentence. Also: You mention that the distribution of IgG subtypes is not clearly elucidated, and you state in line 57, that this is the reason to analyze IGHG3. For me, this argumentation is not conclusive. This raises the question why you only measured IgG3 and not IgG1, 2, 4....
Answer) Thank you for the comment. We studied IgG3 levels in different body fluids because we realized salivary IGHG3 levels were higher in patients with SLE among numerous proteins in 2DE proteomic analysis as we discussed the results deeply in Discussion. We added the rationale to measure and compare IGHG3 levels in different body fluids and underlined them in the revised manuscript.
- Hypergammaglobulinemia is a common manifestation and IgG-immune complex deposition in target tissues results in organ damage in SLE. However, distribution of IgG subclass in the sera and tissues and their respective functions have not been clearly elucidated in lupus. Previous our data showed salivary IGHG3 levels were higher and associated with disease activity markers in patients with SLE [4].
- line 64: please rewrite the sentence like this: "The mean (or median?) age of the patients donating saliva, serum and urine was .....". Otherwise, it does not make much sense. also: please mention also the mean or median age of the healthy controls. Regarding the question if mean or median should be displayed in this manuscript: Please test for normal distribution. But in the methods section, you mention that you used Mann Whitney test for significance. So you assume that the data is not normal distributed. It would be better to show median and range for all the data...
Answer) Thank you for the comment. We applied the Quantile-quantile plots and the Shapiro-Wilk's test, and found that basic characteristics had normal distribution, so we presented as Table 1. But one subgroup among the subgroups divided by clinical features did not follow a normal distribution, so we used the Mann Whitney test. Also, the Spearman correlation was used to determine the strength of association between IGH3G and other variables because IGH3G is not normal for all datasets.
- line 88: ROC curves. Please describe in the method section how you produced the ROC curves. There are different methods to do this.
Answer) Thank you for the comment. But we don’t understand which explanation and what different methods. We analyzed by using SPSS software, and Youden’s index was used to determine the cutoff values as described in the manuscript.
- line 93: ng or μg? please use the same unit.
Answer) Thank you for the comment. Concentration of IGHG3 in saliva is measured as ng/mL, concentration of IGHG3 in serum is measured as μg/mL, and concentration of IGHG3 in urine is measured as ng/mL.
- line 157-167: I am not sure, why this description of IgG3 structure and polysaccharide binding is relevant. As I understand it, it is used as a potential marker in the paper. The link to SLE is not clear. Can you describe more, why the structure and binding characteristics are important for SLE and is there an idea, why IGHG3 is elevated in SLE?
Answer) Thank you for the comment. The described actions of IgG3 in Line 162 – 174 were associated with pathogenesis of SLE. The involvement of IgG3 into complement activation, modulating type I IFN response of IgG3, the interaction of IFNγ and IgG3 on B cells, and IgG3 production from different classes of B cells of lupus-prone mice are not complete, but currently revealed link of IgG3 and SLE. Moreover, there were several studies about IgG3 in pathologic findings on SLE, which are described in next phrase. We modified and added some sentences to inform that those findings are associated with SLE.
- line 226: can you say how many patients you have who provided all 3 specimen or 2? Would it be interesting to show correlated data on this small cohort?
Answer) Thank you for the comments. About 60 patients with SLE provided all 3 specimens, and 80 patients provided 2 specimens. We could not get any statistic power from them, because the number was too small to analyze a correlation.
- line 269: what is "the unstimulated method"? please explain briefly.
Answer) Thank you for the comment. There are the stimulated and unstimulated method in the research of oral diseases or Sjogren’s disease. We added the brief explanation as below and underlined in the revised manuscript.
- There are two methods, the stimulated and unstimulated way, to collect saliva in clinical practice or research. In here, the saliva was collected using the unstimulated method, which is no use of any stimulator.
- Discussion: The value of this marker for diagnosing active nephritis is quite clear. Can you discuss, what benefit IGHG3 has over other parameters (C3 levels, anti-dsDNA, SLEDAI) for the assessment of disease activity and for diagnosis? How do you propose to use IGHG3 in the future?
Answer) Thank you for the comment. We modified the sentence in line 225-235, added our opinion about use of IGHG3 in the future as below and underlined them in the revised manuscript.
- Urinary IgG3 levels correlated with the hemoglobin level, C3 level, ESR, anti-dsDNA antibody positivity and SLEDAI-2K score, suggesting that higher urinary IgG3 levels indicate an active status of SLE or active LN. Urine sampling is non-invasive and much easier than blood sampling, which is mandatory in management of SLE. Urinary IgG3 cannot be a substitute for several disease activity indicators in blood test, but it might be helpful to differentiate exacerbation of SLE and other diseases in situation of fever or skin rash.
- Moreover, urinary IgG3 showed a remarkable value in diagnostic active LN. Although more research is needed, urinary IgG3 test could be used in checking the presence of active LN. Many patients with active LN present with only asymptomatic proteinuria, resulting in delayed intervention and organ damage. It might be considered beneficial to add IgG3 level to proteinuria in regular urinalysis.
Reviewer 2 Report
The manuscript by Jung et al. is a research article aimed to investigate the levels of IGHG3 in saliva, serum, and urine in patients with SLE and verify whether they are associated with disease activity and renal involvement.
A few questions/comments for the authors.
1. In how many patients was IGHG3 measured in serum, urine and saliva at the same time? It is unusual that there is quite a big difference in the age of patients in the "saliva" and "serum" group compared to the "urine" group, and the difference between the "saliva" and "urine" groups is 12 patients.
2. What is the reason that a SLEDAI-2K score greater than 6 was taken as a cut-off between patients with active and inactive disease?
3. Is it possible to examine the influence of drugs on IGHG3 levels, especially glucocorticoids and immunosuppressants?
4. Is there a possibility to examine the association between IGHG3 levels in urine and proteinuria or albuminuria?
5. Is it possible to investigate the association of IGHG3 levels in urine with certain elements in the pathohistological findings of kidney biopsies?
6. The authors should emphasize that they used log odds ratio.
Author Response
Dear Reviewer,
We appreciate your review of our manuscript “Immunoglobulin Gamma-3 Chain C in plasma, urine, and saliva in systemic lupus erythematosus”. In response to your comments and those of the reviewers, we have made several changes to the text, as summarized below.
- In how many patients was IGHG3 measured in serum, urine, and saliva at the same time? It is unusual that there is quite a big difference in the age of patients in the "saliva" and "serum" group compared to the "urine" group, and the difference between the “saliva" and "urine" groups is 12 patients.
Answer) Thank you for the comments. About 60 patients with SLE provided all 3 specimens, and 80 patients provided 2 specimens. Therefore, we considered the study groups of saliva, serum, and urine as 3 different cohorts.
- What is the reason that a SLEDAI-2K score greater than 6 was taken as a cut-off between patients with active and inactive disease?
Answer) Thank you for the comments. In statistical analysis, researcher should decide it according to the previous data. We decided SLEDAI_2K > 6 which had been used in many studies. (Závada J, et al. Serum tenascin-C discriminates patients with active SLE from inactive patients and healthy controls and predicts the need to escalate immunosuppressive therapy: a cohort study. Arthritis Res Ther. 2015 Nov 25;17:341, Tselios K, et al. CD4+CD25highFOXP3+ T regulatory cells as a biomarker of disease activity in systemic lupus erythematosus: a prospective study. Clin Exp Rheumatol. 2014 Sep-Oct;32(5):630-9)
- Is it possible to examine the influence of drugs on IGHG3 levels, especially glucocorticoids and immunosuppressants?
Answer) Thank you for the comments. We added drug data including glucocorticoids and hydroxychloroquine, but the uses of specific drug or dose of glucocorticoids were not associated with IGHG3 levels.
- Is there a possibility to examine the association between IGHG3 levels in urine and proteinuria or albuminuria?
Answer) Thank you for the comments. Most patients with active lupus nephritis have proteinuria. We thought it would be more accurate to compare with the presence of pathological diagnosis (ISN/RPS class III, IV, or V) rather than the presence of uncertain proteinuria.
- Is it possible to investigate the association of IGHG3 levels in urine with certain elements in the pathohistological findings of kidney biopsies?
Answer) Thank you for the comments. No, it is not possible. Our data didn’t contain other elements in the pathohistological findings of kidney biopsies.
- The authors should emphasize that they used log odds ratio.
Answer) Thank you for the comments. We corrected them in revised manuscript.
Thank you for the constructive review. We hope that the revised manuscript now meets the journal’s standards for publication.